# Fabrication and Characterization of a Lead-Free Cesium Bismuth Iodide Perovskite through Antisolvent-Assisted Crystallization

**DOI:** 10.3390/nano14070626

**Published:** 2024-04-02

**Authors:** Salma Maneno Masawa, Chenxu Zhao, Jing Liu, Jia Xu, Jianxi Yao

**Affiliations:** 1Beijing Laboratory of Energy and Clean Utilization, North China Electric Power University, Beijing 102206, China; salma.masawa@udom.ac.tz (S.M.M.); zhaochenxu@ncepu.edu.cn (C.Z.); 18811257982@163.com (J.L.); xujia@ncepu.edu.cn (J.X.); 2Department of Petroleum and Energy Engineering, College of Earth Sciences and Engineering, The University of Dodoma, Dodoma 41218, Tanzania; 3State Key Laboratory of Alternate Electrical Power System, North China Electric Power University, Beijing 102206, China

**Keywords:** all-inorganic perovskite, lead free, antisolvent, cesium bismuth iodide, solar cell

## Abstract

Cesium bismuth iodide perovskite material offers good stability toward ambient conditions and has potential optoelectronic characteristics. However, wide bandgap, absorber surface roughness, and poor surface coverage with pinholes are among the key impediments to its adoption as a photovoltaic absorber material. Herein, bandgap modification and the tailoring of surface morphology have been performed through molar ratio variation and antisolvent treatment, whereby type III antisolvent (toluene) based on Hansen space has been utilized. XRD and Raman spectroscopy analyses confirm the formation of a 0D/2D mixed dimensional structure with improved optoelectronic properties when the molar ratio of CsI/BiI_3_ was adjusted from 1.5:1 to 1:1.5. The absorption results and Tauc plot determination show that the fabricated film has a lower bandgap of 1.80 eV. TRPL analysis reveals that the film possesses a very low charge carrier lifetime of 0.94 ns, suggesting deep defects. Toluene improves the charge carrier lifetime to 1.89 ns. The average grain size also increases from 323.26 nm to 444.3 nm upon toluene addition. Additionally, the inclusion of toluene results in a modest improvement in PCE, from 0.23% to 0.33%.

## 1. Introduction

Lead-based perovskite thin film solar cell absorbers have demonstrated a significant improvement in power conversion efficiency (PCE) from 3.8% to 26.1%, which is now comparable to that of single-junction crystalline silicon-based solar cells (26.1%) [1,2]. However, silicon-based solar cells account for 95% of the worldwide market, while thin-film perovskite solar cells have yet to enter the commercial sector owing to concerns about lead stability and toxicity [3,4,5,6]. Therefore, improving stability and replacing lead with a less toxic substance are the primary objectives of contemporary perovskite solar cell research. Tin (Sn), bismuth (Bi), antimony (Sb), copper (Cu), and germanium (Ge) have been identified as potential lead substitutes [4,7]. Solar cells using organic–inorganic hybrid lead-free tin-based perovskite absorbers now have a PCE of more than 15%, while all-inorganic tin-based lead-free perovskites have a PCE greater than 10% [6,8,9,10,11,12]. This advancement in PCE paves the way for perovskite thin film material toward overcoming the lead toxicity challenge [13,14,15,16]. Moreover, perovskites made with all-inorganic lead-free absorbers have demonstrated superior stability in the face of humidity, light, and heat [13,17,18,19]. All-inorganic lead-free perovskites solar cells created by chemically combining Cs, Bi, and halide elements are among the lead-free compounds under investigation [20,21,22,23]. When compared to tin-based perovskites which offer the highest efficiency among lead-free substitutes, cesium bismuth iodide provides superior stability under ambient conditions. Sn-based perovskites as well as Ge-based perovskites are very unstable upon air exposure due to self-doping. Sn^2+^ and Ge^2+^ normally oxide to Sn^4+^ and Ge^4+^ and overcoming their oxidation necessitates the application of several additive engineering techniques, which increases the cost of solar cell production [24,25,26,27]. This is in contrast to Bi-based perovskite films which form a protective layer of Bi_2_O_3_ or BiOI upon exposure to air which protects the film from environmental degradation [28,29,30,31].

However, despite its diversity in application, its non-toxicity, and its stability superiority, the research focal point on cesium bismuth iodide perovskite materials is directed toward improving the film’s surface morphology and bandgap adjustment with regard to its modest performance as a photovoltaic solar absorber material. The limited solubility of CsI and BiI_3_ precursors in organic solvents such as DMF or DMSO, along with the fast crystallization rate, has a negative impact on the quality of the film’s morphology. The surface deteriorates, leaving numerous voids and pinholes, as well as having poor reproducibility. Moreover, low dimensionality causes the materials to have a large bandgap, increased exciton energy, low charge carrier mobility, and modest absorption intensity [20,32]. Nevertheless, uniform surface morphology and good surface coverage are important film formation parameters for the production of perovskite cells with good optoelectronic properties and high reproducibility [6]. Antisolvent engineering techniques have been extensively deployed to tailor the quality of film morphology [33,34,35].

Antisolvents act as heterogeneous catalysts, speeding up the nucleation rate by increasing the rate of instantaneous supersaturation during the spinning process [36]. Supersaturation occurs upon antisolvent addition due to a reduction in solubility of the precursor solvents and the solvent washing away [35]. The type of antisolvent, the ratio between the precursor solvent and the antisolvent, and the antisolvent dripping time and speed are among key parameters to be considered for enhanced efficiency [37]. Even though the interaction between the solvent and antisolvent or between the precursor and antisolvent is governed by the nature of the antisolvent, an ideal antisolvent is required to be insoluble with both the host solvent and perovskite precursors but also miscible with host solvent [38]. When chlorobenzene was utilized in the fabrication of bismuth-based perovskite films, the result was a decrease in grain size and a modest improvement in surface coverage [39]. Ghosh and coworkers performed a comparative study on the use of antisolvents on a lead-free cesium bismuth iodide perovskite for solar cell applications. Type I (isopropanol), II (chlorobenzene), and III (toluene) antisolvents were used. According to Hansen space perspectives, type I antisolvents are alcohol-based and therefore operate on the basis of hydrogen bonding. Type III possesses a low contribution from hydrogen bonding with negligible polar forces and is almost immiscible with the common types of solvents such as DMF and DMSO, while type II receives contributions from both. Upon utilization with cesium bismuth iodide solar cells, type III antisolvents (toluene) offered the highest power conversion efficiency of 0.046%, followed by chlorobenzene (0.04%), and the lowest PCE was achieved with isopropanol (0.02%) [40]. The influence of several kinds of antisolvents, including methanol, toluene, isopropanol, diethyl ether, chlorobenzene, and chloroform, on the crystallization of CsBi_3_I_10_ perovskite film was investigated [41]. The produced film was observed to dissolve in diethyl ether, chloroform, and methanol, leading to a low absorption intensity and poor quality of film morphology. It was discovered that the CsBi_3_I_10_ film performed better with toluene, chlorobenzene, and isopropanol, with chlorobenzene exhibiting outstanding results in film-quality tailoring. The optimized solar cell device displayed a PCE of 0.63% [41]. Toluene was further used to enhance the crystallinity of bismuth-based perovskite solar cells using a single or double A-site cation; the single A-site cation-based bismuth perovskite achieved a PCE of 0.24% with toluene as the antisolvent, while with the double A-site cation, the PCE was 1.5% [42]. With regard to bandgap adjustment, several techniques have been put in place, such as through compositional engineering as well as molar ratio variation. Adjusting the molar ratio from 1.5:1 to 1:3 resulted in the fabrication of the first cesium bismuth iodide (CsBi_3_I_10_) with a bandgap of 1.77 eV and improved its optoelectronic properties. Johanson further performed a comprehensive investigation on molar ratio variation by considering several ratios of CsI, 1:1, 1.5:1, 1:2, 1:3, and 1:9, and BiI_3,_ and discovered that any compound fabricated with a molar ratio of 1.5:1 containing a mixture of CsI and BiI_3_ possesses improved optoelectronic properties [29,43]. Our group reported an improved morphological quality when cesium bismuth iodide was fabricated with a precursor ratio of 1:1.5 (CsI/BiI_3_) via solvent vapor annealing. A PCE boost from 0.23% under conventional annealing to 0.98% upon DMF/CH_3_0H solvent vapor annealing was reported [44]. However, this cesium bismuth iodide compound fabricated at a 1:1.5 ratio has not yet been fully explored and most information regarding its electronic properties is unknown.

Herein, we report the morphological, structural, and optical characterization of a cesium bismuth iodide perovskite film fabricated at a molar ratio of (CsI/BiI_3_) 1:1.5. We further applied antisolvent-assisted crystallization with toluene as the antisolvent to enhance the quality of film morphology through a one-stage approach as shown in Figure 1. XRD and Raman spectroscopy analyses confirmed the formation of a cesium bismuth iodide with a mixed 0D/2D dimensionality structure. The absorption data and Tauc plot determination showed that the fabricated film has a lower bandgap of 1.80 eV than that of Cs_3_Bi_2_I_9_ (2.03–2.30 eV) at a molar ratio of 1.5:1. The average grain size also increased from 323.26 nm to 444.3 nm upon toluene addition. A modest increase in PCE from 0.23% to 0.33% was also observed upon toluene addition.

## 2. Materials and Methods

### 2.1. Materials

Bismuth iodide (BiI_3,_ 99%), 4-tert-butylpyridine (TBP), tris (2-(1H-pyrazol-1-yl)-4-tert-butylpyridine-cobalt (III)Tris(bis(trifluoromethylsulfonyl)imide)) (FK 209), Bis(tri-fluoromethane) sulfonimide lithium salt (99.95%, Li-TFSI, trace metals basis), and chlorobenzene (99.8%, anhydrous) were obtained from Sigma Aldrich (St. Louis, MO, USA). Acetylacetone (C_5_H_8_O_2_, 99.0%) and Tetra-isopropyl orthotitanate (C_12_H_28_O_4_Ti, s.g 0.97) were from TCI (Tokyo, Japan). Acetonitrile, Ti0_2_ paste 30NR-D, and cobalt (III)tris(trifluoromethylsulfonyl)imide) were from Greatcell solar materials (Queanbeyan, NSW, Australia). Cesium iodide (CsI, 99.99%) was obtained from Acros (Geel, Belgium), while ethanol absolute (99.9%) was from Innochem (Pyeongtaek, Republic of Korea). Isopropanol, methanol, dimethyl sulphoxide (DMSO, anhydrous, ≥99.9%), and N, N-dimethylformamide (DMF, anhydrous, 99.8%) were obtained from Alfer Asar (Ward Hill, MA, USA). 2,2,7,7-Tetrakis (N, N-p-dimethoxyphenylamino)-9,9-spirobifluorene (spiro-OMeTAD) Spiro OMeTAD was from Xi’an Polymer Light Technology Corp (Xi’an, China). All chemicals and reagents were used without further purification.

### 2.2. Perovskite Film and Solar Cell Fabrication Process

The glass substrates coated with fluorine-doped tin oxide (FTO, Pilkington, TEC 15, Lathom, UK) were ultrasonically cleaned with detergent, deionized water, and ethanol for 20 min each prior to ozone treatment. The compact layer was then formed using spray pyrolysis, and the mesoporous electron transport layer was further deposited using the spin coating method. The steps for preparing the solution and for deposition are the same as previously reported [44]. A one-step toluene-assisted crystallization method was used to fabricate the perovskite solar cell absorber. The film was fabricated using CsI and BiI_3_ at a molar ratio of 1:1.5 dissolved in a mixture of DMF and DMSO at a volume ratio (900/100). The molar ratio of 1:1.5 was adopted instead of 1.5:1 since molar ratio variation has been previously reported to alter the optoelectronic properties of cesium bismuth iodide perovskite absorbers. The antisolvent dripping time was varied between 10 s and 15 s and the volume was adjusted from 150 mL to 200 mL. The optimum dripping time and volume was found to be at the 15th s and 200 mL, respectively. The perovskite precursors were dissolved in 1 mL of the solution mixture of DMF and DMSO at 900:100 (*v*/*v*). The hot perovskite precursor solution (70 °C) was spin-coated for 30 s on top of a mesoporous layer at a speed of 3000 rpm. In the last ten to fifteen seconds of the spinning process, about 200 µL of toluene was applied on the antisolvent-treated substrates. The films were then annealed at 70 °C for a few minutes and then at 140 °C for 30 min. After the annealing process, the 2,2,7,7-Tetrakis (N, N-p-dimethoxyphenylamino)-9,9-spirobifluorene (spiro-OMeTAD) was spin-coated at a speed rate of 4000 rpm for 20 s. The spiro-OMeTAD precursor solution was prepared by dissolving 73.5 g of spiro-OMeTAD in 1 mL of chlorobenzene with 29 µL of TBP, 17.0 µL of a Li-TFSI solution, and 8 µL of an FK209–cobalt (III) TFSI solution. The Li-TFSI solution and cobalt (III) solution were prepared by dissolving 520 mg of Li-TFSI salt and 300 mg of FK209–cobalt (III) TFSI salt, respectively, in 1 mL of acetonitrile. We deposited a 70 nm to 80 nm thick gold electrode using the thermal evaporation method at a pressure of 4 × 10^−4^ MPa. Then, a black mask defined at the 0.0627 cm^2^ active layer was used for characterization.

### 2.3. Characterization

A UV–vis spectrophotometer (UV-2450, Shimadzu corp, A 10834735110, Kyoto, Japan) was used to examine the film’s UV–vis spectra. The XRD patterns were recorded by using an X-ray diffractometer (XRD, Smart Lab, Rigaku, Tokyo, Japan) with Cu-Kα radiation (1.5418). The scanning of diffraction angles ranged from 10° to 60° at a scanning speed of 5° per min. Scanning electron microscopy (SEM) images were obtained using an SU8010 SEM (Hitachi, Matsuda, Tokyo, Japan). The current density vs. voltage (J-V) characterization was examined on a fabricated perovskite solar cell with the configuration of FTO/c-TiO_2_/m-TiO_2_/CBI/spiro-OTAD/Au using a Keithley 2400 (Cleveland, OH, USA) source-meter together with a sunlight simulator (XES-300T1, SAN-EI Electric, AM 1.5 G 100 Mw/cm^2^, Tokyo, Japan). Standard silicon was used as a reference cell. The photoluminescence (PL) and time-resolved photoluminescence (TRPL) spectra on the glass substrates were measured using an FS5 Spectrofluorometer (Edinburgh Instruments, Oxford, UK). The Raman spectra measurements were carried out using a diode laser with a 532 nm light source. X-ray photoelectron spectroscopy (XPS) measurements were carried out using an RBD upgraded PHI-5000C ESCA system (Perkin Elmer, Waltham, MA, USA) with MgKᾳ radiation. hµ = 1486.6 eV and the contact angle were obtained using Dataphysics OCA15EC (Filderstadt, Germany). A multimode atomic force microscope (AFM), namely, an Agilent Technologies 5500 instrument (Santa Clara, CA, USA), was used to assess the surface roughness in tapping mode.

## 3. Results and Discussion

### 3.1. XRD Analysis

Cesium bismuth iodide perovskite was fabricated using a mixture of DMF and DMSO (9/1) (*v*/*v*) with and without toluene treatment. DMF was selected due to its strong dissolution power, while DMSO was chosen due to its strong coordination ability [45]. The molar ratio of CsI/BiI_3_ was adjusted from 1.5:1 for Cs_3_Bi_2_I_9_ to 1.15 since adjustments in the precursor molar ratio cause significant improvements in the optoelectronic properties. X-ray diffraction (XRD) measurements were carried out on the fabricated CBI perovskite at diffraction angles ranging from 10° to 60° to evaluate the influence of toluene on the crystallinities of the perovskite films. The XRD patterns of both the control and toluene-treated samples exhibited comparable diffraction peak positions, as shown in Figure 2, implying that the addition of toluene does not lead to structural changes. The highest peaks were found to be at the (101) and (202) planes, with corresponding angles of 12.8° and 25.8°, indicating (101) preferred orientation. However, there are diffraction peaks of (113) and (300) planes at 26.97° and 41.56°, respectively, in both the control and toluene-treated films. These peaks indicate that BiI_3_ is present in the crystal lattice. The main characteristic peaks for BiI_3_ that have been documented in the literature are found in planes (113) and (300) [46,47]. The results validate the existence of hybrid 0D-Cs_3_Bi_2_I_9_ and 2D BiI_3_ structures. A change in the molar ratio leads to the creation of a mixed-dimensional structure [48]. Moreover, all other diffraction peaks correlate well with the fabricated cesium bismuth iodide perovskite films reported in the literature and there are no peaks that can be assigned to CsI crystals; this signifies the full consumption of cesium iodide [29,44].

### 3.2. Ultraviolet–Visible Absorption Analysis

The ultraviolet–visible absorption spectrum analysis in Figure 2b shows that the toluene-treated sample does not exhibit a discernible increase in absorption intensity. According to the absorption data in Figure 2c from the Tauc plot analysis, the bandgap of the cesium bismuth iodide photovoltaic absorber fabricated at a ratio of CsI/BiI_3_ (1:1.5) is determined to be 1.80 eV, which is lower than that of the well-reported Cs_3_Bi_2_I_9_ film fabricated at 1.5:1, which has a bandgap between 2.20 eV and 2.03 eV. Therefore, altering the molar ratio results in a modification of the crystal structure, which in turn enhances the optoelectronic capabilities.

### 3.3. Raman Spectroscopy Analysis

We further performed Raman spectroscopy analysis to confirm the formation of cesium bismuth iodide perovskite film and study the exciton–phonon interactions. Raman spectroscopy has been identified as an efficient experimental method to analyze the exciton–phonon interactions [49]. Both the control film and the toluene-treated film exhibited characteristic peaks at 146.5 and 113.9 cm^−1^. The wavenumber at 146.5 cm^−1^ corresponds to the terminal Bi-I symmetric stretch, while the characteristic peak at 113.9 cm^−1^ is not assigned to any stretching or vibrational mode with regard to cesium bismuth iodide but is the strongest characteristic peak for BiI_3_ crystal which corresponds to the intense zone center C(G) phonon mode of Ag’ symmetry [49,50,51,52]. The findings imply that both the control and toluene-treated films contain a mixture of two phases, 0D-Cs_3_Bi_2_I_9_ and 2D-BiI_3_. Other minor characteristic peaks were also observed at 71 cm^−1^ which originated from BiI_3_ crystal in the control film, which shifted to 61 cm^−1^ upon toluene addition The wavenumber at 61 cm^−1^ is the characteristic peak coming from Cs_3_Bi_2_I_9_ crystal. Moreover, the toluene-treated sample was found to have stronger peak intensity compared to the pristine one, as shown in Figure 2d.

### 3.4. Scanning Electron Microscopy Analysis and Atomic Force Microscope Analysis

Scanning electron microscopy (SEM) analysis was further performed to investigate the effect of toluene addition on the film morphology of CBI perovskite. It can be clearly seen in Figure 3a,b that there is an increase in perovskite crystal size and a reduction in the number of pinholes in the toluene-treated sample. The mean crystal size has increased from 323.3 nm to 444.3 nm upon toluene addition. Both the toluene-treated film and the control film have a mixture of hexagonal and irregular grains. Many pinholes affect the coverage of subsequent layers such as the hole transport layer and metal contact layer, hence limiting the power conversion efficiency. Increased crystal size is advantageous since large grains tend to have a lower density of the grain boundaries and, as a result, a lower number of the corresponding defects located in the grain boundaries. A film containing a high concentration of small grains or grain boundaries is susceptible to Shockley–Read–Hall (SRH) recombination. The grain boundaries serve as non-radiative recombination centers since they are vulnerable to point defect formation, serving as deep traps. An et al. reported that the smallest grain size is linked to a short charge carrier lifetime, while large grains have a longer charge carrier lifetime [53]. The trap density was found to increase linearly with decreasing grain size and it has even been discovered that charge recombination at the grain boundary is significantly more potent than it is at the film surface without surface passivation [53,54,55]. Other film microstructure properties such as the size and density of pinholes, crystallite orientation, and crystallinity quality should also be clearly controlled to achieve a high PCE since nanoscale inhomogeneity and morphological defects are the underlying causes of limited PCE. A decrease in defect density results in an enhancement in charge carrier mobility and a higher diffusion length, which are directly linked with cell power conversion efficiency.

Figure 3c,e show AFM images of the control film with low quality, consisting of irregular, small-sized, randomly distributed grains with voids, confirming the consequence of fast-induced crystallization. When toluene is added, shown in Figure 3d,f, the grains grow in size, revealing a hexagonal lattice crystal with a large number of small-sized irregular grains. The density of pinholes has been reduced, but they can still be visible after adding toluene, indicating that full film coverage has not been attained. There was no improvement in root-mean-square surface roughness seen with the control (77.5 nm) or toluene-treated films (118 nm). The increase in surface roughness may be due to uneven increase in grain size in the target film.

Toluene, as schematically depicted in Figure 4, has a tendency to drive the perovskite into a metastable zone instead of a supersaturated zone. In the metastable zone, there is consistency and uniformity in the nucleation and crystal growth rate, leading to the formation of a perovskite film with large grains and a low grain boundary [56].

### 3.5. X-ray Photoelectron Spectroscopy Analysis

In order to further study the chemical state of Cs, Bi, and I and the interaction between toluene and cesium bismuth iodide perovskite elements, we performed X-ray photoelectron spectroscopy (XPS) measurements. A slight decrease in binding energy from 738.8 eV to 738.6 eV and from 724.9 eV to 724.7 eV can be clearly observed for Cs 3d in Figure 5a upon toluene addition. The shifting of peaks signifies the presence of interaction between toluene and cesium bismuth iodide perovskite. The slight blue shift signifies a modest change in the chemical environment created upon toluene addition. The shift might be caused by the chemical nature of the neighboring atoms on the cesium bismuth iodide surface. The presence of higher valence anionic (Bi_2_I_9_)^3−^ octahedron units which are not bound to each other causes the electron density to increase, and consequently, a decrease in binding energy is observed.

Moreover, for Bi-4f, the characteristic peaks of Bi4f_7/2_ and Bi4f_5/2_ are found at 158.9 and 164.2 eV, respectively. Furthermore, on both pristine and toluene-treated films, further peaks have been identified at 162.6 eV after Bi-4f_5/2_ and 157.2 eV after Bi-4f_7/2_. These peaks are caused by the existence of metallic Bi^0^, which is the primary factor causing poor device performance as a result of induced recombination [51,58]. No shifting of peaks has been observed on the Bi-4f scan and I scan, meaning that toluene addition has an effect on cesium only; this indicates that toluene addition does not cause significant changes in the electronic state and hence will have an insignificant contribution to the overall power conversion efficiency.

### 3.6. FTIR Spectra and Ultraviolet Photoelectron Spectroscopy Analysis

Fourier transform infrared (FTIR) spectra were obtained in order to investigate the effect of toluene on the surface chemistry of cesium bismuth iodide perovskite film, as shown in Figure 6a–c. The peak at wavenumber 1531.7 cm^−1^ belongs to the functional groups of (S=O) stretching bonds from DMSO, while the peak at 621.4 cm^−1^ which lies in the fingerprint region is ascribed to the C-S functional group. Upon toluene addition, the peaks have shifted from 1531.7 cm^−1^ to 1528.05 cm^−1^ and from 621.4 cm^−1^ to 619.4 cm^−1^. The shifting of peaks indicates more robust binding strength between toluene, CsI, and BiI_3_ complexes.

### 3.7. J-V Characterization, and PL and TRPL Analysis

Time-resolved photoluminescence (TRPL) analyses were also carried out to examine the charge recombination behavior of the CBI perovskite films fabricated with and without toluene treatment (Figure 7a). The two exponential decay curve fittings, displayed in Appendix A, reveal that the charge carrier lifetime of the pristine film is prolonged from 0.94 ns to 1.53 ns upon toluene addition.

This modest improvement in carrier lifetime signifies the ability of toluene to tailor the quality of film morphology, hence reducing trap-assisted surface or bulk recombination and, consequently, non-radiative recombination. Improved crystallinity, which results in a lower defect density, is indicated by an increase in grain size and a notable decrease in density and pinhole size. This is reflected in an improvement in the charge carrier lifetime and PCE. Therefore, a longer charge carrier lifetime denotes better crystallization and less non-radiative recombination activity. An increase in charge carrier concentration, absorption intensity, and absorption wavelength corresponds to a decrease in bandgap and an improvement in electron transport efficiency, which affects current density, FF, and Voc, as well as the device’s overall performance.

However, in general, this charge carrier lifespan is still too short, indicating the possibility of CBI film to still have deep defects that act as recombination sites, resulting in a reduced lifetime. The photoluminescence peak positions for the pristine and toluene-treated films were almost identical and were around 710 nm. However, there was an obvious increase in absorption intensity with toluene-treated sample, as shown in Figure 7b.

The photovoltaic performance of the cesium bismuth iodide perovskite solar cell with a configuration of FTO/c-TiO_2_/m-TiO_2_/CBI/Spiro-OMeTAD/Au was further analyzed. The current density–voltage (J-V) characteristics study under standard AM 1.5 G illumination is shown in Figure 7c. The pristine film’s power conversion efficiency increased from 0.24 to 0.33% upon toluene addition. Moreover, the open-circuit voltage, Voc, slightly decreased from 0.45 V to 0.43 V, while the current density, J_sc_, and fill factor increased from 1.63 to 2.14 mA/cm^2^ and from 33.15% to 36.98%, as shown in Table 1. Dark-state J-V measurements were further carried out to estimate the charge accumulation and extraction capabilities at the device interfaces for both pristine and toluene-treated perovskite solar cells. Under forward bias, the toluene-treated perovskite solar cell had a lower dark current and a longer extraction slope than the pristine solar cell (Figure 7d). This indicates that the use of toluene promotes efficient charge extraction capabilities while reducing charge accumulation.

## 4. Conclusions

In summary, we have fabricated and characterized a low-bandgap cesium bismuth iodide perovskite solar cell using toluene-assisted crystallization. The fabricated perovskite film was found to have a bandgap of 1.80 eV and an extended absorption wavelength to 710 nm. Toluene addition regulated the crystallization rate, which enabled the advancement in the quality of film morphology. The perovskite average grain size grew from 323.22 nm to 444.3 nm. Perovskite films with large grains are advantageous due to the reduced grain boundary defect density, hence minimizing trap-assisted surface or bulk recombination and, consequently, non-radiative recombination.

As a result, an increase in absorption intensity, phase purity, charge carrier lifetime, and PCE, from 0.24 to 0.33%, have been attained. However, it should be noted that the modest performance may be contributed by other factors such as energy mismatch between electron/hole transport materials. This research study paves the way for future research on electron and hole transport material optimization and other methods for tailoring surface morphology, such as passivation techniques.

## Figures and Tables

**Figure 1 nanomaterials-14-00626-f001:**
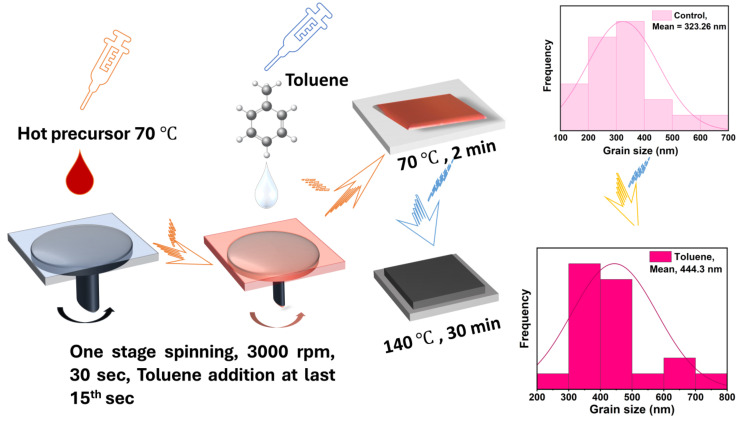
Fabrication of cesium bismuth iodide with toluene as antisolvent and morphological advancement through grain size enlargement.

**Figure 2 nanomaterials-14-00626-f002:**
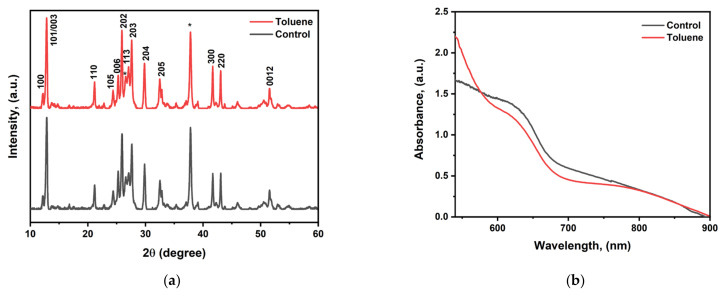
(**a**) XRD pattern for cesium bismuth iodide fabricated with and without toluene as antisolvent. (**b**) UV–visible spectrum for control and toluene-treated samples. (**c**) Bandgap determination using Tauc plot analysis. (**d**) Raman spectra for control and toluene-treated perovskite films (* stands for SnO_2_ from FTO).

**Figure 3 nanomaterials-14-00626-f003:**
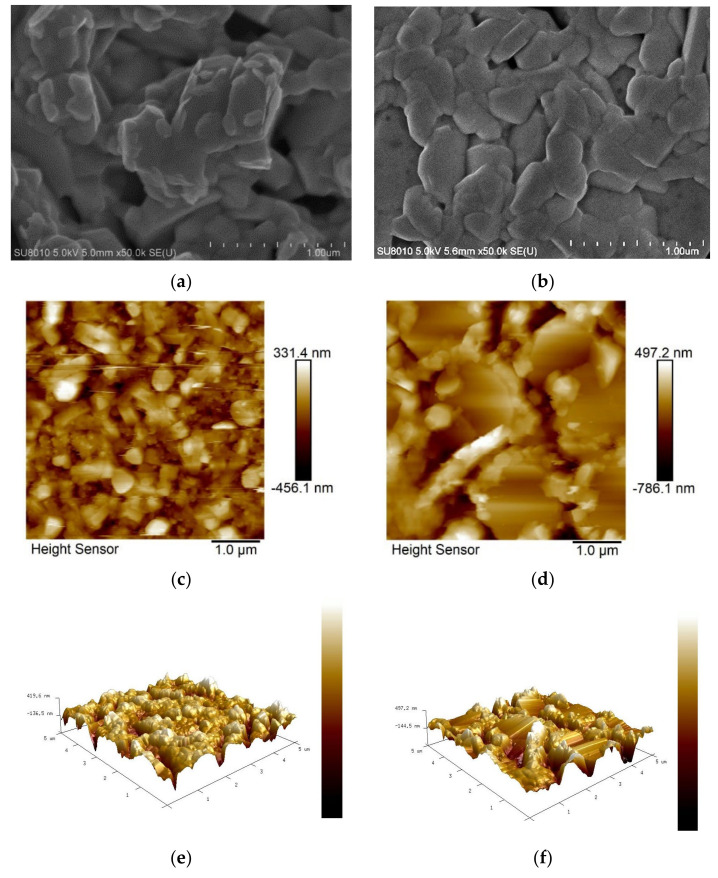
SEM and AFM analyses for (**a**,**c**,**e**) control and (**b**,**d**,**f**) toluene-treated samples, respectively.

**Figure 4 nanomaterials-14-00626-f004:**
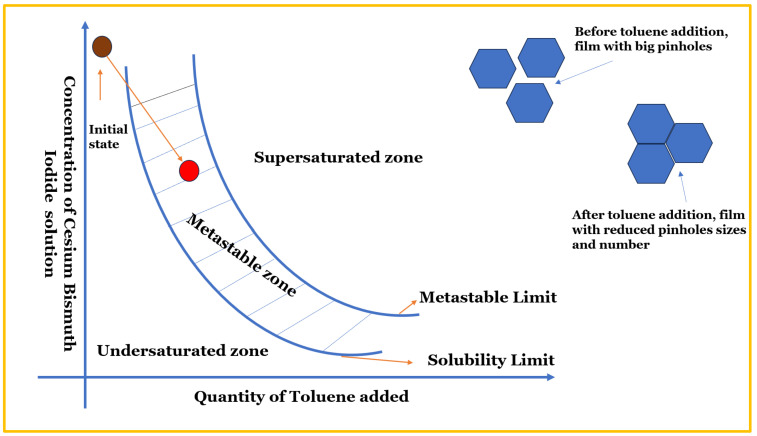
A schematic diagram showing the toluene tendency to shift the perovskite from the under-supersaturated to metastable zone, hence enhancing film morphology [57].

**Figure 5 nanomaterials-14-00626-f005:**
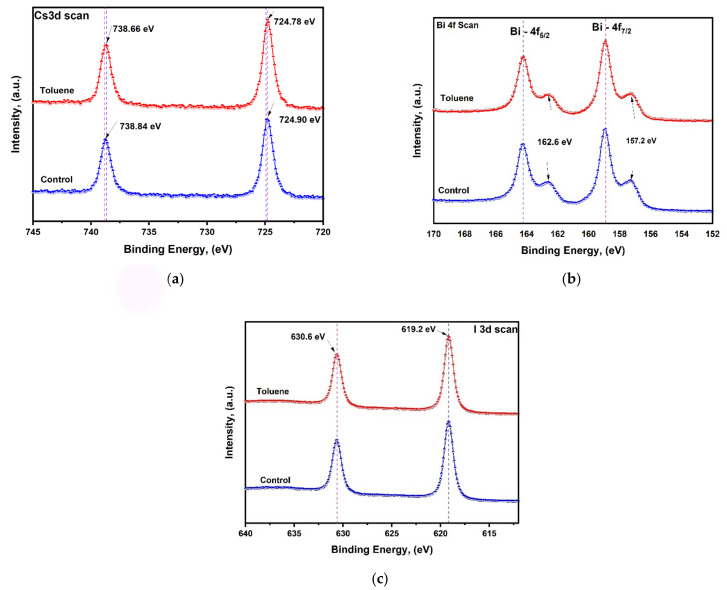
XPS analysis for control and toluene-treated samples on (**a**) Cs scan, (**b**) Bi scan, and (**c**) iodine scan.

**Figure 6 nanomaterials-14-00626-f006:**
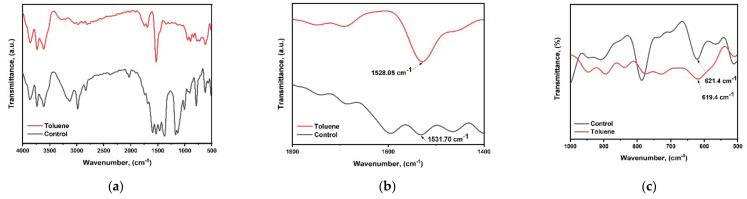
FTIR analysis (**a**–**c**) for control film and toluene-treated film.

**Figure 7 nanomaterials-14-00626-f007:**
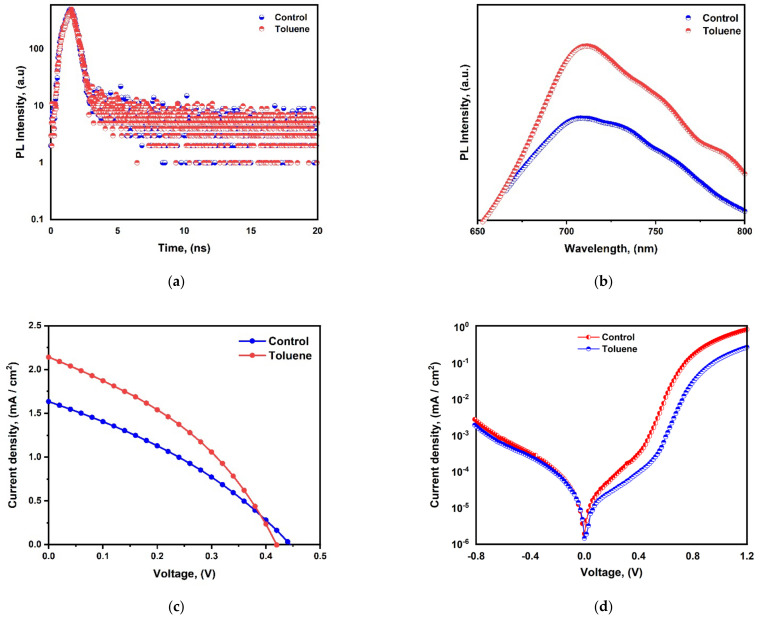
(**a**) Time-resolved PL intensity; (**b**) PL intensity; (**c**) J-V characteristics; (**d**) J-V Characteristics under dark current measurements.

**Table 1 nanomaterials-14-00626-t001:** J-V Characteristics Results.

	Scan	Voc (V)	Jsc (mA/cm^2^)	FF (%)	PCE (%)
Control sample	R_v_	0.45	1.63	33.15	0.24
	F_w_	0.45	1.62	32.17	0.23
Toluene-treated sample	R_v_	0.42	2.14	36.98	0.33
	F_W_	0.43	2.13	34.42	0.32

## Data Availability

Data will be available upon request from the corresponding author.

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
