# Peer review of "Fabrication and Characterization of a Lead-Free Cesium Bismuth Iodide Perovskite through Antisolvent-Assisted Crystallization"

_nanomaterials, 2024, doi:10.3390/nano14070626_

Round 1

Reviewer 1 Report

Comments and Suggestions for Authors

The authors report the morphological, structural and optical characterization on a cesium bismuth iodide perovskite film modified by assisted crystallization with toluene.  Despite the modesty increases in PCE upon toluene addition, this report opens the way to further methods for tailoring surface morphology for electron and hole transport materials optimizations.

Author Response

Dear reviewer, Thank you so much for timely response and well constructive comments. Kindly receive the word document.

Reviewer 2 Report

Comments and Suggestions for Authors

Comments on the Quality of English Language

Can be improved. 

Author Response

Dear, reviewer, thank you so much for timely feedback and very constructive comments. Kindly receive the response.

Round 2

Reviewer 2 Report

Comments and Suggestions for Authors

The authors have addressed all the comments and can observe significant improvement in the manuscript, showcasing more detailed investigation and novelty in the understanding of this research.

Comments on the Quality of English Language

Minor English polishing